# Functional and ecomorphological evolution of orbit shape in mesozoic archosaurs is driven by body size and diet

Stephan Lautenschlager [1✉]

The orbit is one of several skull openings in the archosauromorph skull. Intuitively, it could be assumed that orbit shape would closely approximate the shape and size of the eyeball resulting in a predominantly circular morphology. However, a quantification of orbit shape across Archosauromorpha using a geometric morphometric approach demonstrates a large morphological diversity despite the fact that the majority of species retained a circular orbit. This morphological diversity is nearly exclusively driven by large (skull length > 1000 mm) and carnivorous species in all studied archosauromorph groups, but particularly prominently in theropod dinosaurs. While circular orbit shapes are retained in most herbivores and smaller species, as well as in juveniles and early ontogenetic stages, large carnivores adopted elliptical and keyhole-shaped orbits. Biomechanical modelling using finite element analysis reveals that these morphologies are beneficial in mitigating and dissipating feeding-induced stresses without additional reinforcement of the bony structure of the skull.

[1] School of Geography, Earth & Environmental Sciences, University of Birmingham, Birmingham, UK.  ✉email: s.lautenschlager@bham.ac.uk

Vertebrate skulls display several openings and cavities housing a variety of different soft-tissue structures. However, the distribution of these openings is not uniform across the vertebrate tree and the number and position of the cranial fenestration have been used to distinguish major amniote groups, namely anapsids, synapsids, and diapsids[1]. The presence (or absence) of skull openings has further been used as a diagnostic feature in more inclusive groups, such as the antorbital fenestra in archosauriforms[2]. In addition, the functional aspects of the cranial fenestrations and openings have been considered in several studies: for example, the antorbital fenestra and its relation to different soft-tissues[3] but also its biomechanical role[4]; the size of the temporal opening delimiting the available space for the jaw adductor musculature[5,6]; and the functional significance of palatal vacuities in groups such as temnospondyls[7,8].

In particular, the orbit appears to have received considerable attention with much of the existing work focusing on modern mammals. For example, several studies quantified the shape and orientation of the orbit among different mammal clades[9–12] to find correlations with ecological properties such as locomotion and hunting style. Among extinct vertebrates, orbit and sclerotic ring size have further been used to infer activity patterns in fossil synapsids and dinosaurs[13,14] or to investigate the degree of stereovision in theropod dinosaurs[15,16]. However, apart from a few iconic groups (i.e. theropods) orbit shape has largely been ignored in fossil archosaurs and more broadly in archosauromorphs. This is surprising, as these groups displayed a remarkable taxonomic and morphological diversity[17,18] spanning over 150 million years. Archosauromorphs have followed a series of different evolutionary pathways and occupied a variety of ecological niches throughout the Mesozoic[19,20], with the skeleton adapting to specific functional requirements. The cranial skeleton, in particular, is shaped by competing demands and functional trade-offs (e.g. housing and protection of soft-tissues, bite force generation and feeding performance[21,22]). However, it is unclear if that extends to the shape of the orbit as well.

Conventional wisdom would imply that the shape of the orbit closely follows the shape and size of the eyeball resulting in a predominantly circular morphology. While this appears to be true for some groups of dinosaurs (e.g. some coelurosaurs and maniraptoriforms)[23], other groups, including carnosaurs and tyrannosaurs, appear to have deviated from the circular orbit in adopting a anteroposteriorly compressed shape resembling a figure of eight or keyhole morphology[23]. It is likely that orbit shape in theropod dinosaurs, and more broadly in archosaurs, is correlated with functional and ecological properties as in some modern mammal groups[11,12]. However, previous studies on fossil archosaurs were restricted to small sample sizes and failed to find a link between orbit shape and ecological and functional properties and could not identify the mechanisms driving morphological evolution.

Here, I used geometric morphometric analysis (GMM) to characterise orbit shape across Archosauromorpha and throughout the Mesozoic to quantify the morphological diversity and changes thereof through time. Due to the extensive and well-documented fossil record, Archosauromorpha is ideally suited to identify the mechanisms underpinning ecomorphological evolution manifested in the cranial skeleton. Results from the shape analysis were used for the generation of different theoretical models subsequently subjected to biomechanical analysis to test the functional significance of specific orbit shapes.

## Results

**Morphological variation**. The quantification of orbit shape across all Archosauromorpha using two-dimensional landmarks and principal component analysis shows that nearly three-quarters of the variation is described by PC1 (51%) and PC2 (23%) (Fig. 1). The recovered variation is predominantly expressed in the form of the anteroposterior constriction (along negative PC1), the dorsoventral compression (along positive PC1), and the anteroposterior compression (along PC2) of the orbit. Although many of the sampled taxa approach a circular orbit shape (located in the morphospace plot at 0,0), a number of different species in each group expand the morphospace in different directions. The largest variation is found in Dinosauria, in which the variation is predominantly driven by the adoption of a constricted, keyhole- or figure of eight-shaped orbit morphology. A similar pattern, but with a substantially lower degree of variation, is found in Pseudosuchia and Archosauromorpha (Fig. 1). Within Pterosauria, the morphospace expands towards a constricted but tilted orbit shape. In contrast, Crocodylomorpha occupies a comparably small part of the morphospace described by dorsoventrally flattened orbit shapes.

The morphospace occupation throughout the Mesozoic shows considerable variation often triggered by faunal turnovers (Fig. 2a–g). While Pseudosuchia and Archosauromorpha dominate the morphospace in the first half of the Triassic (Olenekian-Carnian), Dinosauromorpha and Dinosauria are restricted to a few early representatives (Fig. 2g). The latter possess large circular orbits, whereas several archosauromorphs (*Fugusuchus hejiapanensis*, *Erythrosuchus africanus*, *Shansisuchus shansisuchus*) and pseudosuchians (*Batrachotomus kupferzellensis*, *Ornithosuchus longidens*) begin to show elliptical and constricted orbit shapes (Fig. 2g). With the decline of (non-archosaur) Archosauromorpha and Pseudosuchia towards the end of the Triassic (Fig. 2f), Dinosauria occupies a steadily increasing part of the morphospace, culminating in the Late Cretaceous (Fig. 2a). However, orbit shape does not substantially change until the Early Jurassic in species such as *Cryolophosaurus ellioti* (Fig. 2e). This morphospace occupation remains relatively constant until a massive expansion in the Late Cretaceous, predominantly driven by large theropods (e.g. *Skorpiovenator bustingorryi*, *Tyrannosaurus rex*) (Fig. 2a). It is noteworthy that this adoption of non-circular orbit shapes occurs convergently in several groups of non-maniraptoriform theropods (Figs. 2, 3).

Parallel to Dinosauria, Pterosauria shows a similar trend in the expansion of morphospace starting relatively small in the Late Triassic (Fig. 2f), reaching its maximum in the Early Cretaceous (Fig. 2b) before declining again in the Late Cretaceous (Fig. 2a). As a second group, Crocodylomorpha expands the morphospace in a similar but smaller fashion from the Early Jurassic to the Late Cretaceous (Fig. 2a–e). In both groups, the older/basal taxa show a circular orbit morphology (e.g. *Eudimorphodon ranzii*, *Sphenosuchus acutus*) before adopting more extreme orbit shapes in derived taxa (e.g. *Istiodactylus latidens*, *Malawisuchus mwakasyungutiensis*).

Although the sample size is restricted to a handful of taxa only, this pattern appears to be replicated through ontogeny. Where juvenile specimens could be sampled, such as for *Tyrannosaurus rex* and *Tarbosaurus bataar*, those show a largely circular orbits early on in ontogeny and develop a more typical keyhole-shaped morphology as adults (Fig. 3a). The same pattern can also be observed in Pterosauria and Archosauromorpha (Fig. 3b).

Mapped Euclidean distances representing orbit shapes onto a composite phylogenetic tree demonstrates an uneven distribution across phylogeny (Fig. 4). Non-circular orbit shapes are predominant in non-maniraptorifom theropods, as well as in some pterosaurs, pseudosuchians (i.e. rauisuchians), with only isolated occurrences in crocodylomorphs, archosauromorphs, and ornithopod dinosaurs.

A quantification of skull size demonstrates that the expansion of the morphospace coincides with an increase in skull length

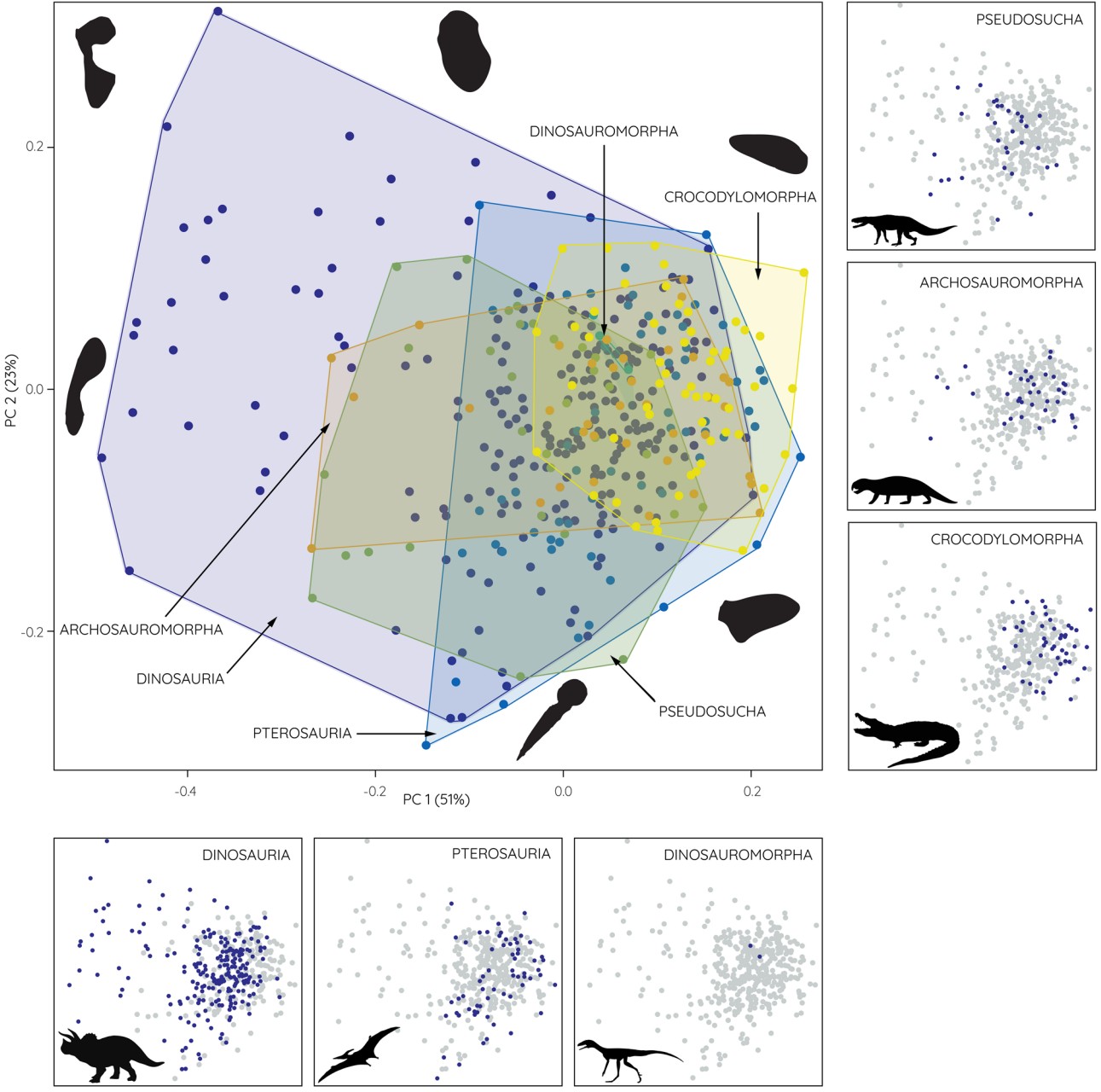

**Fig. 1 Orbit shape morphospace occupation of all archosauromorph taxa (n = 410) and in individual groups.** Silhouettes in the main PCA plot represent extreme orbit shapes.

regardless of taxonomic affinities (Fig. 5a). Similarly, but not exclusively, diet appears to be responsible for some part for variations in orbit shape (Fig. 5b). Although not all carnivorous taxa show deviations from a circular orbit shape, nearly all taxa occupying the negative PC1 (i.e. left) morphospace region representative of a constricted, keyhole-shaped morphology are carnivorous (e.g. theropod dinosaurs) (Fig. 5b).

**Functional results**. A series of different biomechanical analyses were performed to assess the functional impact of different orbit shapes. Subjecting simplified planar models to different compressive and shear scenarios demonstrates that a deviation from the circular orbit morphology can be beneficial in reducing stress concentration under these conditions (Fig. 6). In all tested scenarios simulating dorsoventral and anteroposterior compression and dorsal and anterior shear, keyhole-shaped orbit models

experience reduced stress compared to more circular models (Fig. 6a–d).

Testing a subset of different orbit morphologies in a hypothetical but more realistic skull architecture shows similar results (Fig. 7). While the circular orbit model experiences the highest degree of deformation in a simulated feeding scenario, keyhole- and wedge-shaped orbit models show less deformation. Finite element contour plots of the models further demonstrate that the keyhole- and wedge-shaped orbit configuration redirect stresses away from the jugal and the nasal/frontal region towards the postorbital.

The results from the hypothetical skull models are confirmed in the two *Tyrannosaurus* models (Fig. 8a–f). A hypothetical circular orbit morphology results in a large tensile stress zone along the lacrimal-jugal region (Fig. 8f). In comparison, the original skull/orbit morphology interrupts this stress line

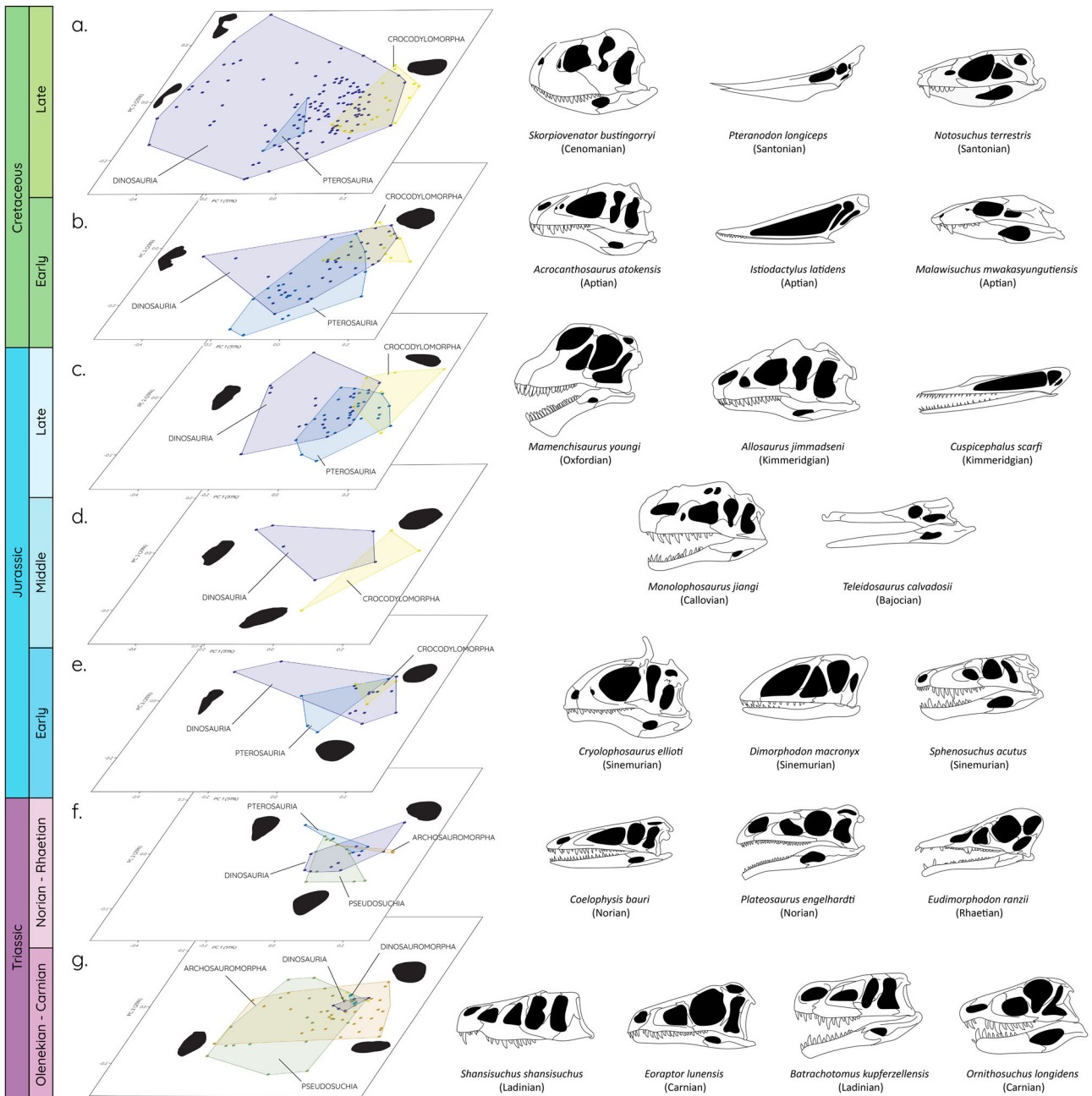

**Fig. 2 Patterns of morphospace occupation through the Mesozoic.** Orbital morphospace plots shown for individual time intervals (**a–g**) and selected taxa representing specific orbit shapes (skull images redrawn based on respective specimen references detailed in the supplementary data 1).

(Fig. 8e) leading to reduced overall von Mises stresses (Fig. 8c, d). Assuming the same muscle mass, the effect on the bite force is negligible with the relative bite force (measured as the ratio between muscle and resultant reaction forces) being 17.37% for the original model and 17.47% for the circular orbit model. The same is true for a reduction of weight for the circular orbit model which has a reduced postorbital bar leading to a decrease in bone volume of 0.5%. However, expanding the eyeball in order to allow it to occupy the maximum available space in both models leads to an increase of over 700% (Fig, 8a, b).

## Discussion
As demonstrated by the results above, and although the majority of species possess orbits that approximate a circular morphology, there is considerable variation in orbit shape between and within different archosauromorph groups. This variation is predominantly driven by large, carnivorous species in each group, suggesting it to be a convergent characteristic independent of phylogenetic affinity (Figs. 4, 5). In particular, species with a skull length of 1000 mm and larger (Fig. 5a) have adopted a keyhole-shaped orbit. This finding confirms results from previous studies conducted on a smaller subset of theropod dinosaurs[23,24]. Both studies also showed that relative orbit size (measured as orbit length compared to skull length and lateral orbit area compared to skull area) decreased in large carnivorous theropods. More basal species and juveniles retaining a circular orbit shape, on the other hand, were found to have relatively larger orbits. These results are reflected in this study here as well and across a considerably larger sample size (Figs. 2, 3). Within each studied

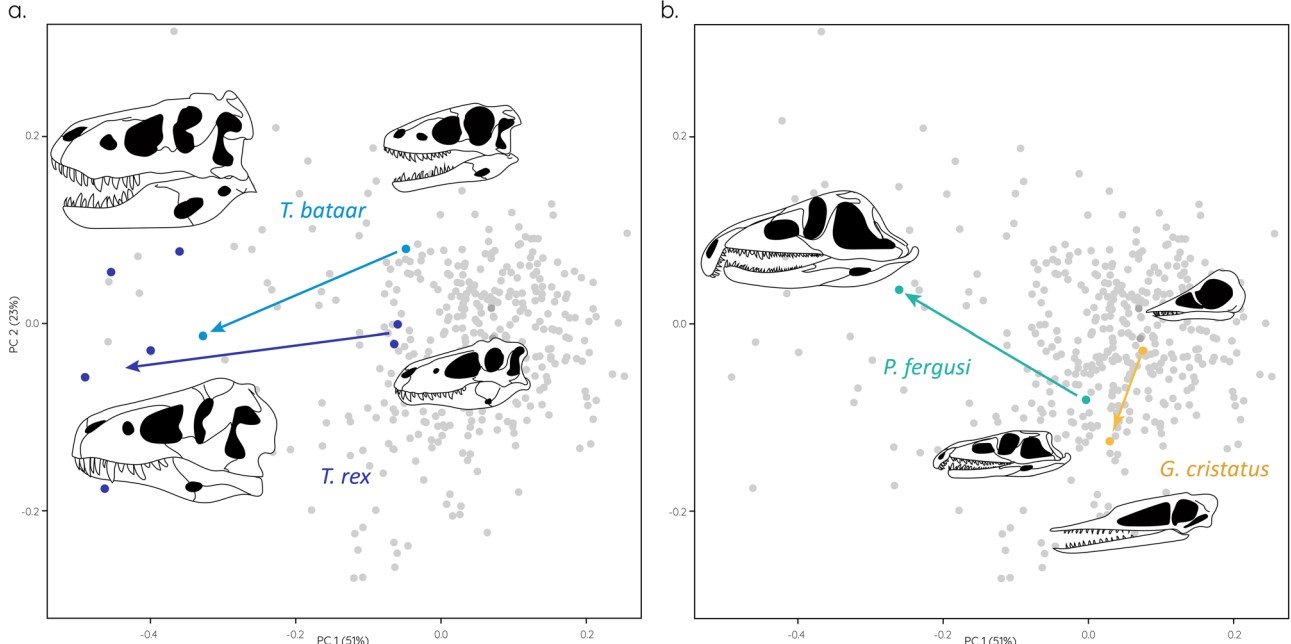

**Fig. 3 Orbital shape changes through ontogeny. a** Juvenile and adult morphospace position and orbit shape for *Tyrannosaurus rex* and *Tarbosaurus bataar*. **b** Juvenile and adult morphospace position and orbit shape for *Proterosuchus fergusi* and *Germanodactylus cristatus*. Morphospace plots as in Fig. 1 (skull images redrawn based on respective specimen references detailed in the supplementary data 1).

archosauromorph group, basal taxa exhibit a circular orbit shape and it is not until further into their evolution and through time that orbital variation increases. Similarly, in species with keyhole-shaped orbits a circular morphology is retained in juveniles further indicating that the development of non-circular orbit shapes is driven by an increase in skull length and a possible change in feeding/hunting behaviour[25].

A reduction in relative orbit size generally also means a reduction in relative eye size, although absolute eye size may be increased in large-bodied species[23]. Furthermore, the deviation from a circular orbit shape reaching its extremes in considerably constricted keyhole- or wedge-shaped orbits restricts the space available for the eyeball even more. As indicated by preserved remains of the sclerotic rings[23] and the progressive constriction of the orbit by projections of the lacrimal and postorbital bones in some species[26], the eye occupied only the dorsal portion of the orbit in these taxa. This is in stark contrast to species with circular orbits for which sclerotic ring and, by inference, eye size could be reconstructed and where the eye occupied a large portion of the orbital cavity[13,27]. Large and well-developed eyes are physiologically expensive and maintaining them may consume up to 15% of an animal's energy budget[28]. As exemplified here for *Tyrannosaurus rex*, retaining a circular orbit occupied completely by the eyeball would result in an increase of eye volume of over 700% (Fig. 8). In vertebrates, increases in eye size can improve visual acuity and sensitivity[29] but the energetic costs of maintaining eyes of this size would most likely outweigh any benefits. Conversely, the reduction in bone material for a circular orbit is negligible (ca. 0.5% or 800 g assuming a specific density of 1900 kg/m³), suggesting that the adoption of non-circular orbit shapes must have provided other functional benefits.

Previous studies have hypothesised that orbit shape was related to feeding biomechanics rather than visual aspects and specifically that elliptical orbits correlated with increased skull strength[23,24]. This hypothesis has been confirmed here in that elliptical, wedge- and keyhole-shaped orbits recorded lower stress concentrations (Fig. 6) and deformation (Fig. 7) in the tested hypothetical models. As

demonstrated for the hypothetical skull models (Figs. 7, 8) circular orbits would increase stresses (in particular tensile stress) in the lacrimal and postorbital bones. The same regions are reinforced in carnivorous theropods[24,30] suggesting that these skeletal elements and their morphological arrangement play a key role in distributing feeding-induced stresses. The presence of (patent) sutures around the orbital bones (i.e. postorbital-jugal and maxilla-jugal contacts) may have acted as further mechanisms to dissipate stresses[31,32]. Henderson[24] further showed that the inclination of the orbit in relation to the long axis of the maxilla has an additional effect in strengthening the skull and that an orientation perpendicular to the maxilla and parallel to the maxillary teeth is advantageous.

Due to the competing demands for space and function changes in orbit shape would lead to other morphological consequences. Circular orbits and the respective eyes occupying them could considerably limit the space of the adductor chamber and thereby the potential muscle volume and redirect muscle paths. Although bite forces are not affected by the change in orbit shape itself as shown here (Fig. 8c, d) a reduction in muscle volume and shallower angles of attack would have a substantial impact on bite forces. In combination with reduced or redirected stresses, non-circular orbits appear to be an adaptation for powerful static biting.

Overall, orbital shape diversity is clearly driven by dinosaurs, and in particular large carnivorous theropods for the reasons outlined above. The morphospace occupation and evolutionary trends of all studied archosauromorphs largely follow the general diversity patterns for these groups. During the Triassic, in the aftermath of the devastating end-Permian mass extinction, archosauromorphs dominated terrestrial ecosystems[33,34]. The end-Triassic mass extinction that saw the extinction of many large archosauromorph groups, such as phytosaurs, rauisuchians, and aetosaurs, however, likely acted as a trigger for the radiation of other groups, such as crocodilians[35] and dinosaurs[36]. During the rest of the Mesozoic, archosaurs continued to diversify, mirroring overall tetrapod diversity patterns[37,38]. However, there are also widespread sampling artefacts documented in the Mesozoic archosaur fossil record[39–42]. Non-archosaurian archosauromorphs and

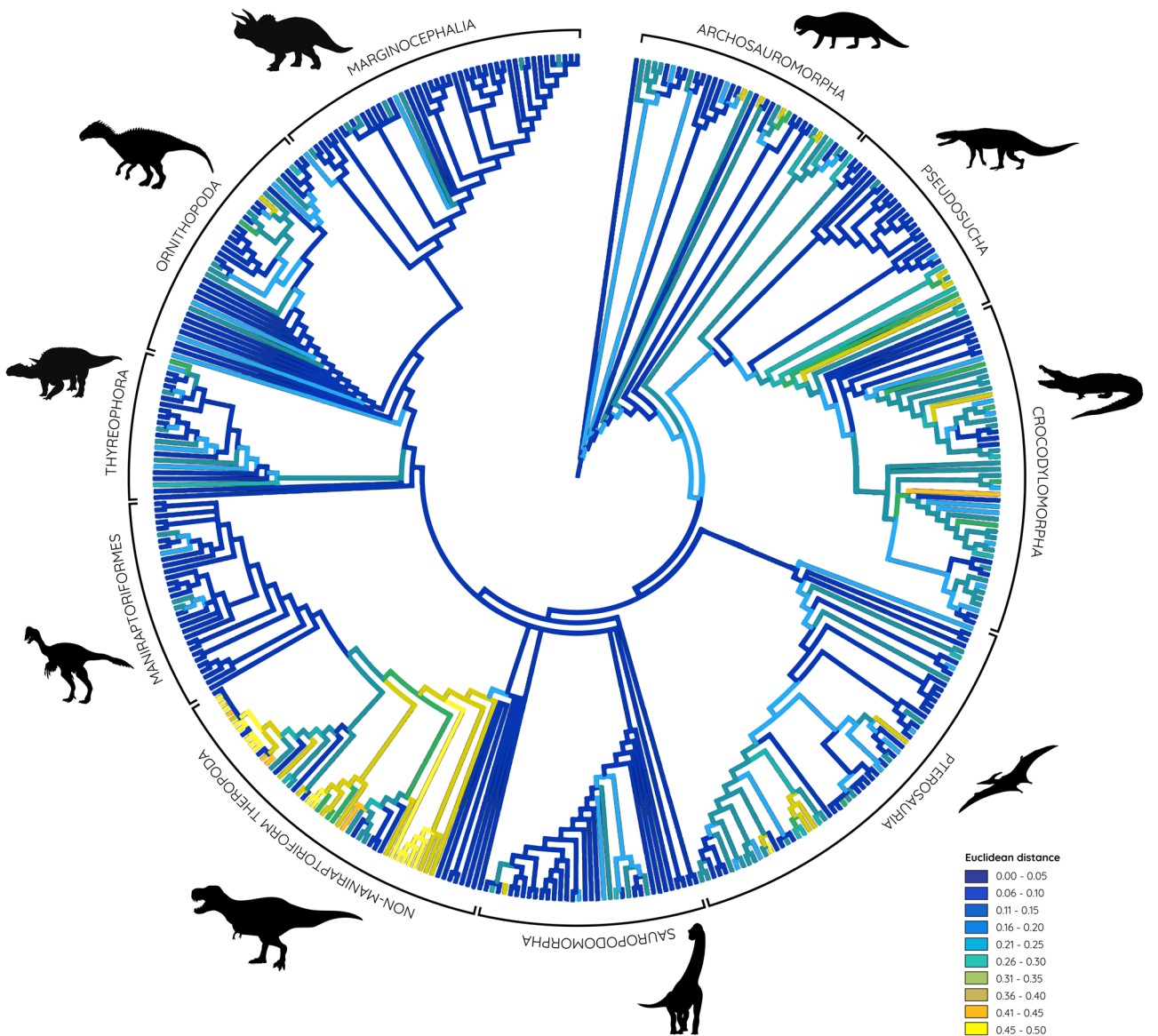

**Fig. 4 Composite phylogenetic tree of analysed species.** Euclidean distances representing different orbit shapes (circular = 0, compressed, keyhole-shape, etc.=0.5) mapped onto phylogeny highlighting occurrences of non-circular orbit morphologies.

pseudosuchians dominate in the Triassic while other archosaurs are largely absent reflecting their origination and subsequent radiation in the Jurassic but also their poor fossil record during the Triassic[43]. The diversity trends in the Jurassic and Cretaceous mirror overall tetrapod diversity patterns in the Mesozoic[38] with an apparent low in the Middle Jurassic. However, this paucity is likely a preservational and sampling artefact[39,40]. Nevertheless, the morphological diversification of the orbit during the Cretaceous appears alongside the emergence of various body plans and morphofunctional adaptations in different archosaurs[20,44,45]. Similarly and consequently, the sampling of taxa here is biased by the availability of fossil skull material and reflected by the uneven distribution for the studied groups. Dinosauria represents over 50% of the sampled specimens, whereas the other groups contribute ca. 10–15% to the overall dataset.

## Conclusions
The quantification of orbit shape across Archosauromorpha shows a wide variety of morphologies but also that the majority of species retained a circular orbit. The morphological diversity is

nearly exclusively driven by large (skull length > 1000 mm) carnivorous taxa and particularly by theropod dinosaurs. This finding parallels the evolutionary trends and diversification of bodyplans and concomitant occupation of ecological niches in dinosaurs more generally. While circular orbit shapes are retained in most herbivores and smaller species, as well as in juveniles and early ontogenetic stages, large carnivores adopted elliptical and keyhole-shaped orbits. These morphologies are beneficial in mitigating and dissipating feeding-induced stresses and require only little investment in reinforcing the bony structure of the skull. Conversely, the development and maintenance of large circular orbits and corresponding eyes would be physiologically costly and likely outweigh potential benefits for visual acuity.

## Methods
**Specimens**. In total, 410 specimens were sampled from the literature (see Supplementary Data 1 for details). Only taxa that preserved the complete orbit were selected, as well as a few incomplete specimens that could be reconstructed with a large degree of confidence. Two-dimensional outlines of the orbit were generated using Adobe Illustrator CC (Adobe Inc.) with the skull orientated so that the maxillary tooth row/the ventral margin of the maxilla was aligned horizontally. For

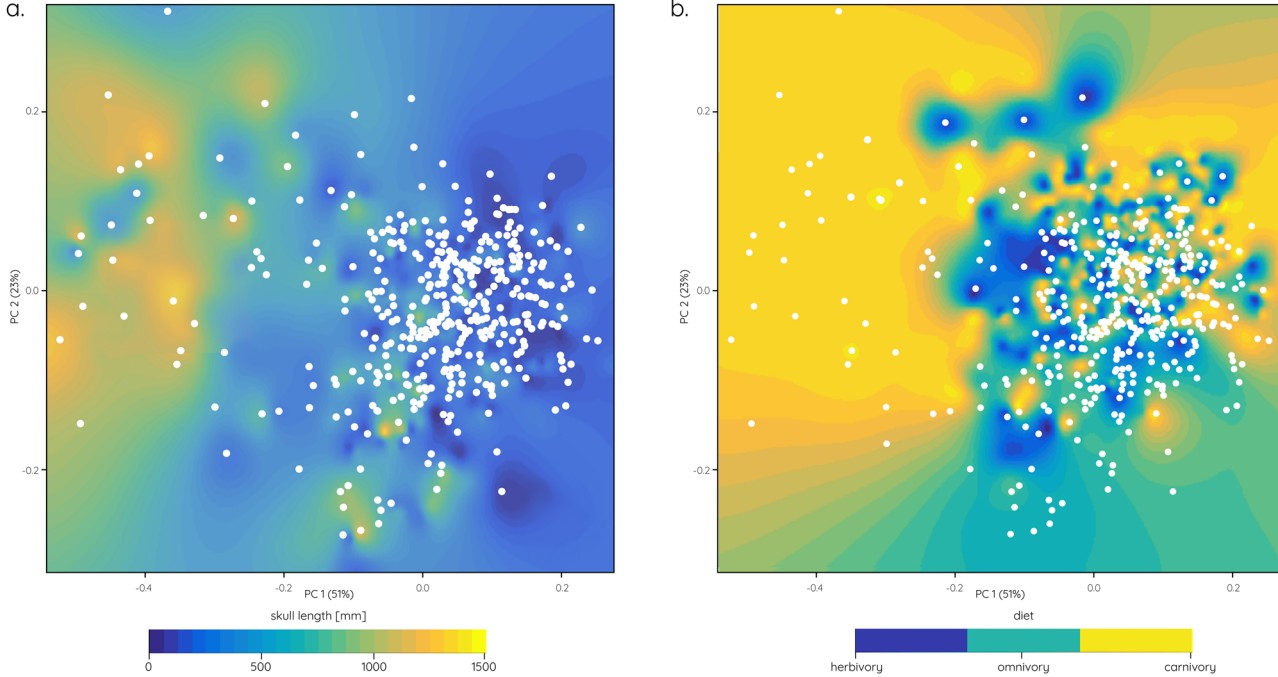

**Fig. 5 Influence of skull size and diet on orbital shape. a** Skull length heatmap superimposed on orbital shape morphospace (as in Fig. 1). **b** Dietary regimes heatmap superimposed on orbital shape morphospace (as in Fig. 1).

some crocodylomorph taxa in which the orbits are facing dorsally the outlines were collected in dorsal view (15 out of 45 taxa; see species list for details). For each specimen, skull length (measured from the tip of the premaxilla or the rostral bone to the occipital condyle), inferred diet (simplified into herbivorous, omnivorous, and carnivorous regimes), and temporal distribution were recorded. In the context of this study, taxa have been grouped based on Ezcurra (2016)[2] and as follows: Archosauromorpha (= all non-archosaurian archosauromorphs), Pseudosuchia (= all non-crocodylomorph pseudosuchians), Dinosauromorpha (= all non-dinosaurian dinosauromorphs), Pterosauria, Dinosauria (all non-avian dinosaurs), and Crocodylomorpha (restricted to Mesozoic taxa).

**Geometric Morphometric Analysis**. Orbit outlines were scaled with the horizontally or vertically largest dimension measuring 1000 pixels. Outlines were then superimposed on a set of two cross-hairs offset at 45 degrees (thereby creating eight equal-sized sectors) and centred horizontally and vertically in Adobe Illustrator to standardise the orientation. The aligned and superimposed orbit shapes were exported as JPEG images for landmarking in tpsDig2[46]. Eight fixed landmarks were selected at the intersections between the orbit outline and the cross-hair lines. Between the fixed landmarks, seven evenly-spaced semi-landmarks were selected for each sector using the curve tool in tpsDig2. This resulted in a total of 64 landmark points with sufficient resolution to describe the entire orbit outline (Fig S1). This approach was taken to standardise the landmarking as the orbit outlines lack homologous points (e.g. osteological features) that could be landmarked consistently for all specimens.

The landmark coordinates were subsequently superimposed using Procrustes analysis and subjected to a principal component analysis (PCA) using PAST 4.03[47]. The PCA scores were used to generate morphospace plots and performance heatmaps (using the R package MBA: https://cran.r-project.org/web/packages/MBA/index.html).

**Phylogeny**. A composite phylogenetic tree was generated to map results from the GMM analysis onto the phylogeny. Data was compiled from Button and Zanno (2020)[48] for Dinosauria, Wang et al. (2008)[49] and Upchurch et al. (2015)[50] for Pterosauria, Bronzati et al. (2012)[51] for Crocodylomorpha, and Nesbitt (2009)[52] and Ezcurra (2010)[2] for Archosauromorpha and Pseudosuchia into a single tree. As orbit shape cannot easily be described as a single character value, Euclidean distances for principal components 1–3 from the GMM analysis were calculated. The centre of the three-dimensional PCA plot (0, 0, 0) represents a circular orbit with shapes deviating the most being characterised by larger Euclidean distances (i.e. positions furthest from the origin). Although this approach does not distinguish between the different types of non-circular orbits it allows mapping them onto the phylogeny. Character mapping was performed in Mesquite 3.7[53].

**Mechanistic models**. In preparation for the functional analyses, three sets of theoretical or mechanistic models were created. The first set consisted of simplified

planar models incorporating different orbit shapes. Planar models with discontinuities (i.e. holes) are classic problem cases in mechanical engineering with the aim to minimise the localisation of high stresses, for example around rivet holes and threads[54,55]. Although this approach simplifies the cranial structure considerably and only considers orbit shape, it allows extracting meaningful performance measures without the compromising influence of other components and cavities in the skull. Therefore, the effects of orbit shape alone can be analysed. For this purpose 21 theoretical orbit outlines covering the entirety of the observed orbit shape diversity were used. These shapes ranged from perfectly circular outlines to elliptical and constricted (i.e. figure of eight) outlines in different orientations (Fig. S2). The dimensions of the planar models were set to a ratio of 10:10:1 (height:width:depth).

The theoretical shapes were landmarked as outlined above and imported into Blender (blender.org, v. 2.83) via a Python script. Landmark coordinates were then transformed into three-dimensional objects using the shrinkwrap modifier. These 3D representations of the orbit shapes were then used to virtually "pierce" a hole in a pre-generated frame model using Boolean operators. Although largely automatic, this process required some manual adjustments to ascertain a correct replication of the orbit shapes.

A second (smaller) set of theoretical models was created to test the effect of orbit shape within a cranial setting. For this purpose, a generic archosaur skull was generated using a box-modelling approach in Blender[56]. The model incorporated a realistic skull architecture, cranial cavities, and relative dimensions and was created with a circular orbit (Fig. S3, S4). In this, the model resembled the general condition found in basal taxa of different archosauriform groups (e.g. Dinosauromorpha, Dinosauria, Pterosauria). Based on the initial model, four further variations were created by changing the orbit manually: (i) an anteroposteriorly compressed elliptical orbit; (ii) a dorsoventrally compressed orbit; (iii) a constricted, keyhole-shaped orbit; and (iv) a wedge-shaped orbit tapering ventrally (Fig. S3).

A third set of hypothetical models was created using an existing model of the Cretaceous theropod dinosaur *Tyrannosaurus rex*. The model was created for a previous study[57] and is based on a cast of BHI 3033 (Black Hills Institute, South Dakota) housed at the Sauriermuseum Aathal, Switzerland. The species was selected as a representative of a large carnivorous archosaur with a keyhole-shaped orbit. In addition to the actual model, a further hypothetical model was generated with a circular orbit in Blender. The hypothetical orbit was created with a diameter that corresponded to the widest dimension of the actual keyhole-shaped orbit to test the biomechanical effect of this morphology (Fig. 8). To estimate the size and volume of the corresponding eyeballs, a digital sphere was virtually placed into the orbit. The sphere was mediolaterally compressed to account for the position of the braincase bones and positioned into the orbit so that the widest dimension was flush with the orbital margins. Orbital musculature was not considered for this simplified spherical reconstruction of the eyeball. Volumes of the skull and eyeball models were directly measured in Blender.

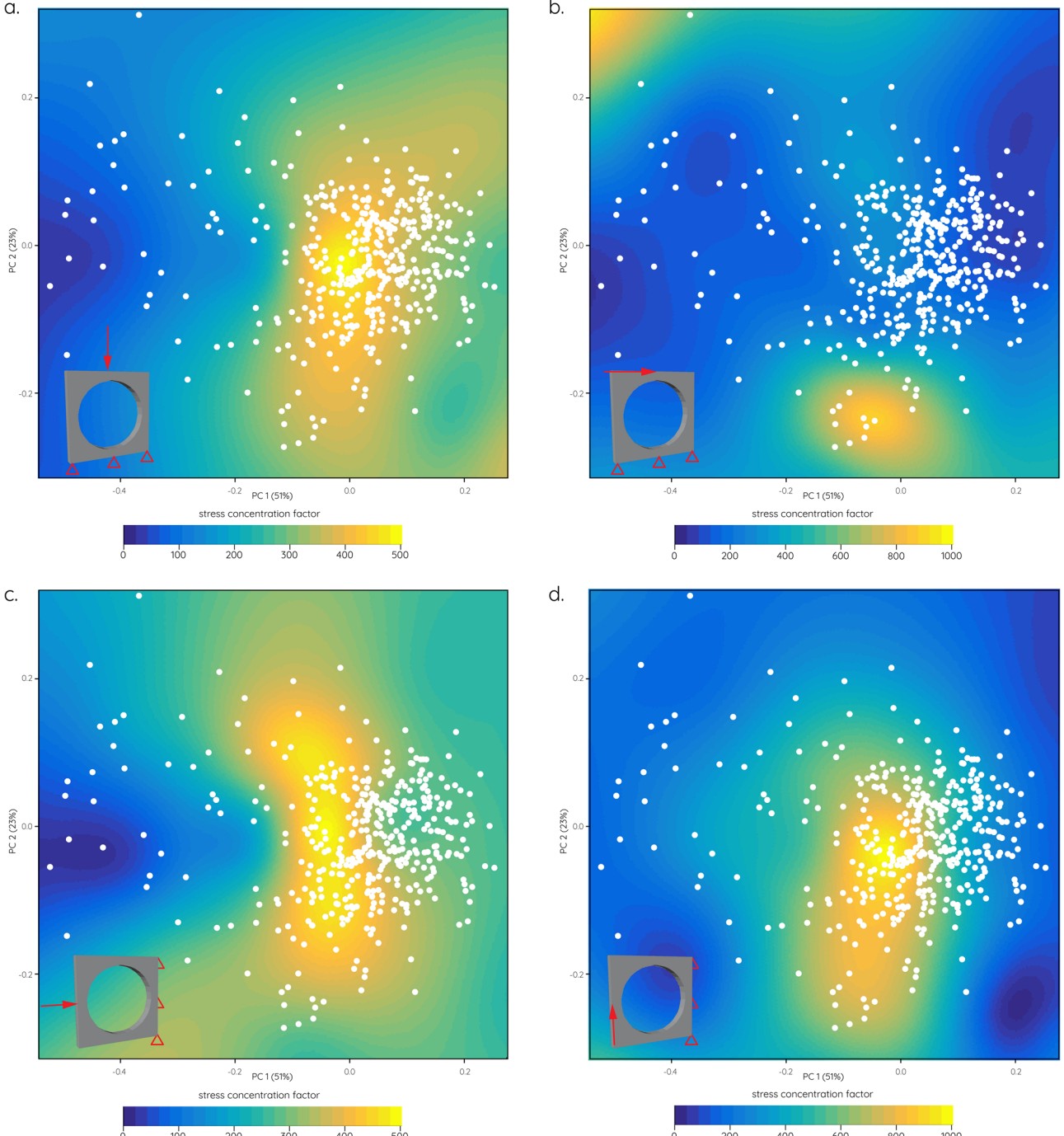

**Fig. 6 Biomechanical performance spaces.** Stress concentration factors (ratio between peak and reference stresses) for tested mechanistic planar models visualised as heatmaps with orbital shape morphospace superimposed. **a** Dorsoventral compression, **b** anterior shear, **c** anteroposterior compression, **d** dorsal shear.

All models were subsequently exported as STL files for the biomechanical analyses.

**Functional analysis**. For the biomechanical analyses, all models generated in the previous step were imported into HyperMesh (Altair, v. 11) for solid meshing and the setting of boundary conditions. Mesh size was kept uniform to generate a quasi-ideal mesh following Marcé-Nogué et al. (2016)[58] which allowed the calculation of average stress values. All models were assigned isotropic material properties for alligator bone as a proxy for archosauromorph bone (E = 15.00 GPa, $\upsilon = 0.29$)[59].

For the flat planar models, four functional scenarios were tested (Fig. 5): (i) dorsoventral compression with the ventral margin constrained at three equidistant points and three ventrally-directed loads placed at the dorsal margin. (ii)

anteroposterior shear with the ventral margin constrained at three equidistant points and three posteriorly-directed loads placed at the dorsal margin. (iii) anteroposterior compression with the posterior margin constrained at three equidistant points and three posteriorly-directed loads placed at the anterior margin. (ii) dorsoventral shear with the posterior margin constrained at three equidistant points and three ventrally-directed loads placed at the anterior margin. A load force of 1 N was selected for each load of the initial circular model. Load forces for all other models were then scaled following the quasi-homothetic transformation approach of Marcé-Nogué et al. (2013)[60] which ensured correct force/surface area ratios.

To test different orbit shapes in a biologically realistic scenario, the hypothetical skull models were subjected to muscle-driven biting. The models were constrained at the quadrates (two constraints in x-, y- and z-direction on each side) and the occipital condyle (four two constraints in x-, y- and z-direction), as well as at an

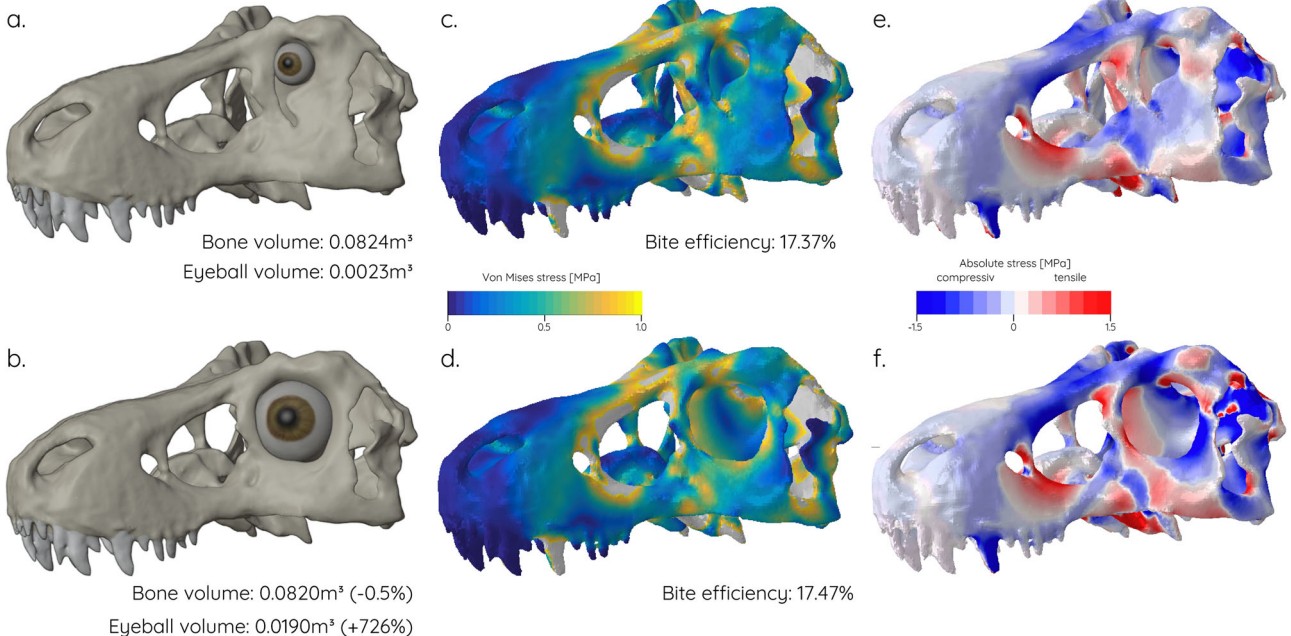

**Fig. 7 Three-dimensional deformation space.** Position of hypothetical skull models visualised for PCs 1-3. Distance between undeformed and deformed models indicated by arrows and calculated Euclidean distances. Von Mises stress contour plots for each model show in undeformed condition.

**Fig. 8 Comparison of actual and hypothetical Tyrannosaurus models. a**, **c**, **e** Original orbit shape, **b**, **d**, **f** circular orbit shape. **a**, **b** Osteological models with reconstructed eyeball fitted to the size of the orbit; **c**, **d** von Mises stress contour plots; **e**, **f** compressive and tensile stress contour plots.

assumed bite point at the seventh tooth (representing an anterior maxillary tooth position) (one constraint in x-, y- and z-direction). To simulate jaw adductor muscles, one load vector with a force of 100 N each created for each of the eight jaw muscle pairs.

For the *Tyrannosaurus* model, a muscle-driven bite scenario was analysed similar to the hypothetical models. Quadrates (four nodes on each side), occipital condyle (eight nodes), and the third maxillary tooth (one node on each side) were constrained from movement in x-, y- and z-direction. The jaw adductor muscles reconstructed and muscle forces were calculated following the protocol outlined in Lautenschlager (2013)[6] to provide realistic load forces for the FEA models (Fig. S5, table S1).

All models were imported into Abaqus (Simulia, v. 6.141) for analysis and post-processing. Biomechanical performance for the flat planar models was assessed by calculating a stress concentration factor. Stress concentrations are formed by the presence of discontinuities (e.g. orbit shapes) in a structure which would otherwise show a uniform stress distribution[61]. A stress concentration factor is a dimensionless metric describing the ratio between peak stress to a reference stress and therefore describes how stress magnitudes are increased due to the introduction to a discontinuity. The reference stress was calculated from the normal stress for the dorsal and anterior compression models and as maximum normal stress for the dorsal and anterior shear models. Peak stresses were obtained from the FE models along the margin or the orbit shape (thereby avoiding artificially high stresses at the constrained nodes).

For the hypothetical skull models biomechanical performance was quantified in the form of model deformation using a landmark-based approach. For the *Tyrannosaurus models* reaction (= bite) forces were used as the main performance quantifier. In addition, contour plots were created to illustrate the figures. For the quantification of deformation, the undeformed and deformed hypothetical models were exported from Abaqus and landmarked in Avizo (Thermo Fisher Scientific, v. 8) (see fig. S4). The landmark data was then subjected to a Procrustes and principal component analysis in PAST. Euclidean distances were calculated to quantify the differences between each model pair (undeformed/deformed).

**Reporting summary**. Further information on research design is available in the Nature Research Reporting Summary linked to this article.

## Data availability
All data (landmarks and phylogeny data for GMM, hypothetical plate and skull models, FEA files) are available here: https://doi.org/10.5061/dryad.1rn8pk0wz[62].

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

## Acknowledgements

Donald Cerio (Johns Hopkins University School of Medicine) and Emma Dunne (University of Birmingham) are thanked for discussion and helpful suggestions which substantially improved this study. Jordi Marcé Nogué (Universitat Rovira i Virgili, Tarragona) and two anonymous reviewers are thanked for constructive feedback on earlier version of the manuscript.

## Competing interests

The author declares no competing interests.
