## [Peer Review File · Communications Biology]

Reviewers' comments:

Reviewer #1 (Remarks to the Author):

See the document attached

Reviewer #2 (Remarks to the Author):

[These comments have also been uploaded in .doc form]

OVERALL COMMENTS -

This manuscript examines the shape of the orbital cavity in a wide range of archosaur-line reptiles. The author notes that the orbit is subcircular in most of the studied taxa, but keyhole-shaped orbital cavities occur in a number of large carnivorous archosaurs. This observation has not been widely noted outside of phylogenetic character matrices, and the study is commendable for its analysis of orbit shape in both a morphospace and functional context. The author constructs a morphospace of orbit shape using 420 skull reconstructions of archosaur-line reptiles. He also constructed a series of idealized skulls with various orbital shapes to examine the effects of various loadings and muscle-driven biting regimes. The results show fundamental changes in the orientation of biting stresses in concert with different orbital shapes. Similar results occur in modified *Tyrannosaurus rex* skull models.

This work is a very interesting examination of orbital shape. Although the orbit contains the eye and associated structures in archosaur-line reptiles, it has not been subjected to a broad analysis like the one found in this manuscript. Prior analyses of cranial cavities in archosaur-line reptiles are usually restricted to the chambers for jaw musculature or pneumatic openings. Ideally, the study would use undistorted archosaur-line skulls for constructing the morphospace plots, but using idealized skull reconstructions expands the sample immensely to the betterment of the work. The morphospace plots corroborate the author's contention that elliptical and keyhole-shaped orbits represent a major departure from the subcircular morphology in most archosaur-line reptiles. I have a few areas of contention with the sample, especially related to the use of playrostral taxa (see below). I cannot comment extensively on the validity of the functional analytical methods due to my limited background in that area, but the authors does present adequate controls and experimental states for the models.

On a large scale, I only have one large criticism of the analytical framework related to the integration of flat-snouted animals like neosuchian crocodylomorphs. The authors restricts his tracing of orbital shape to a lateral view of the skull. However, crocodylomorphs have flattened skulls that fundamentally alter the orientation of the bones surrounding the orbit; they in effect go from being lateral-facing openings to a part of the skull roof. As such, the true orbital cavity of a flat-snouted croc is circular when view dorsally, much like the orbits of other non-hypercarnivorous archosaurs. I'm not sure if there is an easy fix for this particular issue other than eliminating crocs from the analysis or a second run of the analysis in which croc orbits are drawn from dorsal view instead of lateral. I would hypothesize that this would make the divergent nature of big-headed hypercarnivores even more prominent.

I suggest the author track down this paper about the Chinese erythrosuchid *Shansisuchus* as it could be a good addition to the analysis because of the taxon's resemblance to the tyrannosaur/abelisaur orbital morphology:

Wang R, Xu S, Wu X, Li C, Wang S. 2013. A new specimen of *Shansisuchus shansisuchus* Young, 1964 (Diapsida: Archosauriformes) from the Triassic of Shanxi, China. *Acta Geologica Sinica* 87: 1185-1197.

I think the paper's analyses and conclusions are sound, although there are the methodological concerns about incorporating playtrostral crocodylomorphs into the study. There are also a number of small corrections to be made with the dataset, described below.

LINE-BY-LINE EDITS -

Line 10 – closely?

Line 10 - "approximate" seems like a better fit than "represent."

Line 15 – A bit vague. Maybe, “particularly prominently in theropod dinosaurs.”
Line 19 – Not clear what is meant by “investment,” nor clear how that is addressed in the manuscript.
Line 34 - “focusing”
Line 41 - “Archosauromorphs”
Line 45 – Close parentheses.
Line 50 – A sentence or two describing the divergent condition in big theropods would be useful.
Line 52 – Is this reference to previous studies focused on only reptile studies? Vague as written.
Line 68 – Constriction in what plane?
Line 82 – I think further sampling of Erythrosuchidae will show that they are the oldest lineage with this type of orbit.
Line 82 – Should indicate that this refers to non-archosaur archosauromorphs.
Line 87 – Worth noting here that this is multiple different theropod lineages converging on the same shape!
Line 147 – Perhaps “non-circular” rather than “extravagant.”
Line 150 – Perhaps “absolute” instead of “actual.”
Line 196 – Brusatte et al., 2010 requires # reference.
Line 199 – Not clear how this relates to the emergence of body plans, as the theropods with aberrant orbits maintain the body plan of earlier taxa.
Line 212 – Should “likely” be added to the statement “outweight potential benefits...”? That specific hypothesis wasn't tested in the manuscript, although the references about eye size corroborate the idea.
Lines 231-242 – I think a figure illustrating the crosshairs on an example orbit would be valuable. It is hard to visualize based solely on the Methods text.
Line 391 – Reference requires pagination or DOI.
Line 423 – Complete author list.
Line 428 – Complete author list.
Line 445 – Italicize journal and add pagination and/or DOI.
Line 446 – I know it would be cumbersome, but some reference to the actual taxon sampling for each category should go into the figure caption.

DATASET EDITS -

Macrochemus should be Macrocnemus

Hyperodapedon ← was this measure taken from lateral view?

Azendohsaurus laaroussii ← Sure this isn't Azendohsaurus madagaskarensis? Only one known from complete skull materials.

Euparkeria capensis and Osmolskina czatowicensis are currently assigned to Dinosauromorpha.

They should be in Archosauromorpha, as they are non-archosaurian archosauriforms. Euparkeria capensis is actually duplicated, appearing in both sections.

The disparity in the size of the sample used should be more heavily referenced. Dinosauromorpha is so much larger than the others. I think the sampling of taxa is sufficiently thorough, but the temporal and taxonomic depth is much larger than any other group.

Reviewer #3 (Remarks to the Author):

This study provides a comprehensive anatomical description of two-dimensional orbit shape in Mesozoic archosaurs through geometric morphometrics. From my perspective, the study is largely confirmatory in that it corroborates patterns that have emerged through detailed anatomical descriptions. I offer a few major comments:

1) The study is almost entirely descriptive and is not driven by a specific hypothesis or question. It strikes me as somewhat of a “fishing expedition” even though there are existing hypotheses in the literature that could be tested. For example, the hypothesis that elliptical orbit shapes increase skull strength --- that hypothesis represents an opportunity for a detailed analysis. The study offers some data in support of this hypothesis, but the sample size is small. I would also strongly suggest adding more details on muscle anatomy, insertion points to make the models even better;

there is much potential for improvement.

2) An important point to consider is that the "extravagant orbit anatomy" may not have a direct, or immediate function at all, like Gould's spandrels. It is well known from mammals and birds that the eye gets proportionately smaller with eye socket size. In large skulls never fills the orbit completely. This known correlation renders the model in Fig 7b, with a huge eye filling all of the orbit as non-sensible. I would emphasize here again that it would be necessary to carry out more detailed reconstructions of jaw musculature, eyes and eye muscles to better figure out the "soft" anatomy in this region.

3) The 2-dimensional nature of the data is also problematic as the orientation of eye sockets varies substantially across archosaurs. Flattening all orbits into 2 dimensions will distort shapes and can bias the pattern. In this context it should also be noted that the orientation of the eye socket in itself is perhaps more relevant than details of the anatomy of the socket margin itself, as the in-depth studies of mammalian orbits have demonstrated convincingly.

4) The lack of a phylogenetic framework of the analysis is also problematic. It is very widely known that phylogeny is one of the major factors influencing anatomy, and there are many different techniques that can account for phylogenetic covariance, including for geometric morphometrics and PCA. Phylogenies are available and can be time calibrated. Adding phylogenetic context also facilitates much more advanced possibilities to illustrate character evolution.

The presented manuscript tested some functional hypothesis of the orbits in archosaurs that can be of interest to be published in **Communications in Biology**. As an expert in the use of FEA in morphological studies, I validated all the workflows, and my comments and criticisms are based in my expertise in this field. I am not an archosaur specialist, and I did not review the biological part. Most of the work is correctly done and the FEA models are properly built. However, I have two big issues to report which implies a change in the FEA models and FEA inputs that can drive to changes in the discussion of the manuscript. Is for this reason that I am suggesting major revisions.

1) I do not agree with the sentence in line 312 "*The loads do not reflect actual muscle forces as these were irrelevant for the comparative test conducted here*" because I think that the forces are of high importance in these models. I concur with the author that the value is not important for comparative purposes but the proportion between all the muscular forces is important. Applying the same value of force of 100 N in all the eight adductor muscles pairs could drive to absolutely different results if the forces applied in the adductor muscles are defined with heterogeneous values among the eight vector pairs. It will change the distribution of stress and probably the conclusions or justifications of the work. Therefore, I ask the author to use properly related/proportional real values in the muscle forces. There are several ways to modelling them and I am pretty sure that the author will find the best option. If not, I am open to discuss this issue with him.

2) The study of rounded morphologies in structures is a well-known problem that have been addressed specially in aeronautics* and I find specially interesting to develop in biology. So, I applause the idea of the work. Unfortunately, I really think that the use of average values of stress in the plate models is not appropriate. They are not capturing which morphologies are worst designed for handling high stress values because it is well known that the high stress in a window/hole are in the perimeter/outline of the cavity. The average stress will be directly influenced by the amount of elements with low values. This, in a homogeneous mesh is directly related with the surface of plate that remains solid after the creation of the hole. So, I suspect that the morphologies with a big hole (like, for example the rounded hole in the middle of the morphospace) will have a high average value because there are few elements with low values of stress due that under the same force, the stress will be higher in narrow and tiny surfaces. Opposite, the extreme morphologies (the 4 corners of the morphospace) will have more elements with low values because under the same force, there is much more surface to distribute the stress and, consequently, stress will be lower. Therefore, and under my understanding, the results here are biased by the space occupied by the hole and not by the morphology. My suggestion here is studying only the stress in the perimeter/outline of the hole (asking the software the stress in a line), which undoubtedly will capture the effect of the morphology.

*See <https://www.aerotime.aero/23023-why-airplane-windows-are-round>

Some minor points that I suggest reviewing or changing:

1. **Line 32:** I am perfectly aware that Temnospondyls are not archosaurs but, considering that the author is citing in the introduction one work of Temnospondylis that analyse openings in their skulls and points out that orbits have been ignored in fossil archosaurs: Why Marcé-Nogué et al 2015, PLOS One is not cited considering that they already used FEA in Temnospondyls to study the size and position of the orbits?

Marcé-Nogué, J., Fortuny, J., de Esteban-Trivigno, S., Sánchez, M., Gil, L., & Galobart, À. (2015). 3D Computational Mechanics Elucidate the Evolutionary Implications of Orbit Position and Size Diversity of Early Amphibians. PLOS ONE, 10(6), e0131320. <https://doi.org/10.1371/journal.pone.0131320>

2. **Line 230:** I do not understand why the authors are superimposing, centring, and aligning the orbit outlines prior to the Procrustes analysis. The Procrustes analysis includes these steps so, it seems redundant. This redundancy does not affect the analysis and I am wondering if the authors can explain in the manuscript why they do the steps twice. I think that will clarify this part.
3. **Line 246:** "Plate model" is a wrong use of the term. In strength of materials and structures a plate is a flat surface that it is subject to bending. Namely, when forces are applied perpendicular to the surface of the plate. In this case, with surface models with forces lying in the same plane of the model, the right word is "plane models" or "planar models".
4. **Line 254, 269, etc.:** Include which figure of the supplementary information you are citing.
5. **Line 277:** I miss or 1) a citation of the figure 7 where the different morphologies can be seen or 2) a figure in the supplementary information with the geometries with the real and the hypothetical orbits.
6. **Line 291:** I think that a figure of the biomechanical scenarios is needed. Just to understand graphically where the forces and the boundary conditions are applied in a simple look because the written description can be a little cumbersome.
7. **Line 298:** Although the thickness is not relevant for the comparative and the results of stress if you scale the models, I would include a sentence with the thickness you modelled the plates.

Summarizing, I think that the work needs to address the points that can affect the interpretation of the results and, consequently, the discussion before being published.

Jordi Marcé-Nogué

Reviewer #1

The presented manuscript tested some functional hypothesis of the orbits in archosaurs that can be of interest to be published in **Communications in Biology**. As an expert in the use of FEA in morphological studies, I validated all the workflows, and my comments and criticisms are based in my expertise in this field. I am not an archosaur specialist, and I did not review the biological part. Most of the work is correctly done and the FEA models are properly built. However, I have two big issues to report which implies a change in the FEA models and FEA inputs that can drive to changes in the discussion of the manuscript. Is for this reason that I am suggesting major revisions.

Lautenschlager: I would like to thank the reviewer for the constructive and helpful suggestions. This is much appreciated.

1) I do not agree with the sentence in line 312 "*The loads do not reflect actual muscle forces as these were irrelevant for the comparative test conducted here*" because I think that the forces are of high importance in these models. I concur with the author that the value is not important for comparative purposes but the proportion between all the muscular forces is important. Applying the same value of force of 100 N in all the eight adductor muscles pairs could drive to absolutely different results if the forces applied in the adductor muscles are defined with heterogeneous values among the eight vector pairs. It will change the distribution of stress and probably the conclusions or justifications of the work. Therefore, I ask the author to use properly related/proportional real values in the muscle forces. There are several ways to modelling them and I am pretty sure that the author will find the best option. If not, I am open to discuss this issue with him.

Lautenschlager: A fair critique and I agree that the muscle proportions are of course not homogenous in real life as compared to the simplified approach taken here. I have now done a full muscle reconstruction for the species/model based on previously published methods and re-run the analyses with the more realistic muscle values. The changes in the resulting stress distribution are, however, negligible so that the interpretations and inferences from the original version are still valid. In addition, supplementary figure 5 has been added showing the relevant details for the muscle reconstruction.

2) The study of rounded morphologies in structures is a well-known problem that have been addressed specially in aeronautics* and I find specially interesting to develop in biology. So, I applause the idea of the work. Unfortunately, I really think that the use of average values of stress in the plate models is not appropriate. They are not capturing which morphologies are worst designed for handling high stress values because it is well known that the high stress in a window/hole are in the perimeter/outline of the cavity. The average stress will be directly influenced by the amount of elements with low values. This, in a homogeneous mesh is directly related with the surface of plate that remains solid after the creation of the hole. So, I suspect that the morphologies with a big hole (like, for example the rounded hole in the middle of the morphospace) will have a high average value because there are few elements with low values of stress due that under the same force, the stress will be higher in narrow and tiny surfaces. Opposite, the extreme morphologies (the 4 corners of the morphospace) will have more elements with low values because under the same force, there is much more surface to distribute the stress and, consequently, stress will be lower. Therefore, and under my understanding, the results here are biased by the space occupied by the hole and

not by the morphology. My suggestion here is studying only the stress in the perimeter/outline of the hole (asking the software the stress in a line), which undoubtedly will capture the effect of the morphology.

*See <https://www.aerotime.aero/23023-why-airplane-windows-are-round>

Lautenschlager: The reviewer is correct that the results will be dependent on the size of the “remaining” elements. For that reason the load force has been scaled accordingly to accommodate for these differences (lines 330-333).

“A load force of 1 N was selected for each load of the initial circular model. Load forces for all other models were then scaled following the quasi-homothetic transformation approach of Marcé-Nogué et al. (2013)⁵² which ensured correct force/surface area ratios. “

Some minor points that I suggest reviewing or changing:

1. **Line 32:** I am perfectly aware that Temnospondyls are not archosaurs but, considering that the author is citing in the introduction one work of Temnospondylis that analyse openings in their skulls and points out that orbits have been ignored in fossil archosaurs: Why Marcé-Nogué et al 2015, PLOS One is not cited considering that they already used FEA in Temnospondyls to study the size and position of the orbits?

Marcé-Nogué, J., Fortuny, J., de Esteban-Trivigno, S., Sánchez, M., Gil, L., & Galobart, À. (2015). 3D Computational Mechanics Elucidate the Evolutionary Implications of Orbit Position and Size Diversity of Early Amphibians. PLOS ONE, 10(6), e0131320. <https://doi.org/10.1371/journal.pone.0131320>

Lautenschlager: Reference has been added as suggested.

2. **Line 230:** I do not understand why the authors are superimposing, centring, and aligning the orbit outlines prior to the Procrustes analysis. The Procrustes analysis includes these steps so, it seems redundant. This redundancy does not affect the analysis and I am wondering if the authors can explain in the manuscript why they do the steps twice. I think that will clarify this part.

Lautenschlager: This would be correct if fully homologous landmarks would be used. However, given the nature of the orbital outlines it is not possible to use landmarks which are, for example, based on osteological features, thereby providing a consistent framework. Therefore, the approach as outlined in the manuscript has been taken to standardise the landmark collection. A brief explanation has been added (lines 256-257):

“This approach was taken to standardise the landmarking as the orbit outlines lack homologous points (e.g. osteological features) that could be landmarked consistently for all specimens.”

3. **Line 246:** “Plate model” is a wrong use of the term. In strength of materials and structures a plate is a flat surface that it is subject to bending. Namely, when forces are applied perpendicular to the surface of the plate. In this case, with surface models with forces lying in the same plane of the model, the right word is “plane models” or “planar models”.

Lautenschlager: Wording has changed to “planar model” throughout the text.

4. **Line 254, 269, etc.:** Include which figure of the supplementary information you are citing.

Lautenschlager: Figure information has been added.

5. **Line 277:** I miss or 1) a citation of the figure 7 where the different morphologies can be seen or 2) a figure in the supplementary information with the geometries with the real and the hypothetical orbits.

Lautenschlager: Figure reference has been added.

6. **Line 291:** I think that a figure of the biomechanical scenarios is needed. Just to understand graphically where the forces and the boundary conditions are applied in a simple look because the written description can be a little cumbersome.

Lautenschlager: Schematics of the boundary conditions are presented in figure 6 and the figure reference has been added.

7. **Line 298:** Although the thickness is not relevant for the comparative and the results of stress if you scale the models, I would include a sentence with the thickness you modelled the plates.

Lautenschlager: Information has been added (line 285-286):

“The dimensions of the planar models were set to a ratio of 10:10:1 (height:width:depth).”

Reviewer #2

OVERALL COMMENTS -

This manuscript examines the shape of the orbital cavity in a wide range of archosaur-line reptiles. The author notes that the orbit is subcircular in most of the studied taxa, but keyhole-shaped orbital cavities occur in a number of large carnivorous archosaurs. This observation has not been widely noted outside of phylogenetic character matrices, and the study is commendable for its analysis of orbit shape in both a morphospace and functional context. The author constructs a morphospace of orbit shape using 420 skull reconstructions of archosaur-line reptiles. He also constructed a series of idealized skulls with various orbital shapes to examine the effects of various loadings and muscle-driven biting regimes. The results show fundamental changes in the orientation of biting stresses in concert with different orbital shapes. Similar results occur in modified *Tyrannosaurus rex* skull models.

This work is a very interesting examination of orbital shape. Although the orbit contains the eye and associated structures in archosaur-line reptiles, it has not been subjected to a broad analysis like the one found in this manuscript. Prior analyses of cranial cavities in archosaur-line reptiles are usually restricted to the chambers for jaw musculature or pneumatic openings. Ideally, the study would use undistorted archosaur-line skulls for constructing the morphospace plots, but using idealized skull reconstructions expands the sample immensely to the betterment of the work. The morphospace plots corroborate the author's contention that elliptical and keyhole-shaped orbits represent a major departure from the subcircular morphology in most archosaur-line reptiles. I have a few areas of contention with the sample, especially related to the use of playrostral taxa (see below). I cannot comment extensively on the validity of the functional analytical methods due to my limited background in that area, but the authors does present adequate controls and experimental states for the models.

Lautenschlager: I would like to thank the reviewer for the very helpful suggestions and comments. This is much appreciated.

On a large scale, I only have one large criticism of the analytical framework related to the integration of flat-snouted animals like neosuchian crocodylomorphs. The authors restricts his tracing of orbital shape to a lateral view of the skull. However, crocodylomorphs have flattened skulls that fundamentally alter the orientation of the bones surrounding the orbit; they in effect go from being lateral-facing openings to a part of the skull roof. As such, the true orbital cavity of a flat-snouted croc is circular when view dorsally, much like the orbits of other non-hypercarnivorous archosaurs. I'm not sure if there is an easy fix for this particular issue other than eliminating crocs from the analysis or a second run of the analysis in which croc orbits are drawn from dorsal view instead of lateral. I would hypothesize that this would make the divergent nature of big-headed hypercarnivores even more prominent.

Lautenschlager: Apologies if this had not been made clear in the previous version. For crocodylomorphs with a dorsoventral orientation of the orbit the samples have been taken in dorsal view (15 out of 45 species). Other species with an ambiguous orientation have been removed from the analysis now. This is now explained and also indicated in the specimen list.

I suggest the author track down this paper about the Chinese erythrosuchid *Shansisuchus* as it could be a good addition to the analysis because of the taxon's resemblance to the tyrannosaur/abelisaur orbital morphology:

☐ Wang R, Xu S, Wu X, Li C, Wang S. 2013. A new specimen of *Shansisuchus shansisuchus* Young, 1964 (Diapsida: Archosauriformes) from the Triassic of Shanxi, China. *Acta Geologica Sinica* 87: 1185-1197.

Lautenschlager: *Shansisuchus* and *Fugusuchus* have been added to the data set (see also later comment about first appearance of keyhole-shaped orbits).

I think the paper's analyses and conclusions are sound, although there are the methodological concerns about incorporating playtrostral crocodylomorphs into the study. There are also a number of small corrections to be made with the dataset, described below.

LINE-BY-LINE EDITS -

☐ Line 10 –closely?

Lautenschlager: Corrected.

☐ Line 10 -“approximate” seems like a better fit than “represent.”

Lautenschlager: Changed as suggested.

☐ Line 15 –A bit vague. Maybe, “particularly prominently in theropod dinosaurs.”

Lautenschlager: Changed as suggested.

☐ Line 19 –Not clear what is meant by “investment,” nor clear how that is addressed in the manuscript.

Lautenschlager: Changed to the sentence below.

“Biomechanical modelling using finite element analysis reveals that these morphologies are beneficial in mitigating and dissipating feeding-induced stresses without additional reinforcement of the bony structure of the skull.”

☐ Line 34 -“focusing”

Lautenschlager: Corrected.

☐ Line 41 -“Archosauromorphs”

Lautenschlager: Corrected.

☐ Line 45 –Close parentheses.

Lautenschlager: Corrected.

☐ Line 50 –A sentence or two describing the divergent condition in big theropods would be useful.

Lautenschlager: Brief explanation has been added (lines 49-52):

“While this appears to be true for some groups of dinosaurs (e.g. some coelurosaurs and maniraptoriforms)²², other groups, including carnosaurs and tyrannosaurs, appear to have deviated from the circular orbit shape in adopting an anteroposteriorly compressed shape resembling a figure of eight or keyhole morphology²³.”

☐ Line 52 –Is this reference to previous studies focused on only reptile studies? Vague as written.

Lautenschlager: Detail has been added to make the reference clear (lines 54-57):

“However, previous studies on fossil archosaurs were restricted to small sample sizes and failed to find a link between orbit shape and ecological and functional properties and could not identify the mechanisms driving morphological evolution.”

☐ Line 68 –Constriction in what plane?

Lautenschlager: Orientation has been clarified (lines 70-72):

“The recovered variation is predominantly expressed in the form of the anteroposterior constriction (along negative PC1), the dorsoventral compression (along positive PC1), and the anteroposterior compression (along PC2) of the orbit.”

☐ Line 82 –I think further sampling of Erythrosuchidae will show that they are the oldest lineage with this type of orbit.

Lautenschlager: As suggested *Fugusuchus* and *Shansisuchus* have been added to the dataset. Not all Erythrosuchidae show the extremely constricted orbit morphology but do deviate from the circular condition. This is acknowledged now (lines 84-87):

“The latter possess large circular orbits, whereas several archosauromorphs (*Fugusuchus hejiapanensis*, *Erythrosuchus africanus*, *Shansisuchus shansisuchus*) and pseudosuchians (*Batrachotomus kupferzellensis*, *Ornithosuchus longidens*) begin to show elliptical and constricted orbit shapes (Fig. 2g).”

☐ Line 82 –Should indicate that this refers to non-archosaur archosauromorphs.

Lautenschlager: Short-hand definitions of the groups are provided in the methods but the reviewer is of course correct that this may be misleading here as the methods are coming after the results. Corrected accordingly (lines 87-88):

“With the decline of (non-archosaur) Archosauromorpha and Pseudosuchia towards the end of the Triassic (Fig. 2f), Dinosauria occupies a steadily increasing part of the morphospace, culminating in the Late Cretaceous (Fig. 2a).”

☐ Line 87 –Worth noting here that this is multiple different theropod lineages converging on the same shape!

Lautenschlager: Text has been added as suggested and reference to the added phylogeny figure is included (lines 93-94).

“It is noteworthy that this adoption of non-circular orbit shapes occurs convergently in several groups of non-maniraptoriform theropods (Fig. 3).”

☐ Line 147 –Perhaps “non-circular” rather than “extravagant.”

Lautenschlager: Changed as suggested.

☐ Line 150 –Perhaps “absolute” instead of “actual.”

Lautenschlager: Changed as suggested.

☐ Line 196 –Brusatte et al., 2010 requires # reference.

Lautenschlager: Corrected.

☐ Line 199 –Not clear how this relates to the emergence of body plans, as the theropods with aberrant orbits maintain the body plan of earlier taxa.

Lautenschlager: This was not clear on my part. The intended meaning was that the largest diversification of orbit shapes occurs in the Cretaceous, a period during which also different other morphological adaptations in the skeleton appeared rather than those are directly correlated. The sentence has been rephrased to make that clear (lines 210-212).

“Nevertheless, the morphological diversification of the orbit during the Cretaceous appears alongside the emergence of various body plans and morphofunctional adaptations in different archosaurs⁴²⁻³³.”

☐ Line 212 –Should “likely” be added to the statement “outweight potential benefits...”? That specific hypothesis wasn't tested in the manuscript, although the references about eye size corroborate the idea.

Lautenschlager: Fair enough. Added as suggested.

☐ Lines 231-242 –I think a figure illustrating the crosshairs on an example orbit would be valuable. It is hard to visualize based solely on the Methods text.

Lautenschlager: Such a figure is available in the supplementary information (fig. S1).

☐ Line 391 –Reference requires pagination or DOI.

Lautenschlager: Corrected.

☐ Line 423 –Complete author list.

Lautenschlager: Corrected.

☐ Line 428 –Complete author list.

Lautenschlager: Corrected.

☐ Line 445 –Italicize journal and add pagination and/or DOI.

Lautenschlager: No DOI or pagination available for this reference. Correct as it is.

☐ Line 446 –I know it would be cumbersome, but some reference to the actual taxon sampling for each category should go into the figure caption.

Lautenschlager: This had been simplified due to caption length limit.

DATASET EDITS -

☐ *Macrochemus* should be *Macrocnemus*

Lautenschlager: Corrected.

☐ Hyperodapedon ← was this measure taken from lateral view?

Lautenschlager: Yes.

☐ *Azendohsaurus laaroussii* ← Sure this isn't *Azendohsaurus madagaskarensis*? Only one known from complete skull materials.

Lautenschlager: Corrected.

☐ *Euparkeria capensis* and *Osmolskina czatowicensis* are currently assigned to Dinosauromorpha. They should be in Archosauromorpha, as they are non-archosaurian archosauriforms. *Euparkeria capensis* is actually duplicated, appearing in both sections.

Lautenschlager: Corrected.

☐ The disparity in the size of the sample used should be more heavily referenced. Dinosauromorpha is so much larger than the others. I think the sampling of taxa is sufficiently thorough, but the temporal and taxonomic depth is much larger than any other group.

Lautenschlager: This is now acknowledged (lines 212-215).

“Similarly and consequently, the sampling of taxa here is biased by the availability of fossil skull material and reflected by the uneven distribution for the studied groups. Dinosauria represents over 50% of the sampled specimens, whereas the other groups contribute ca. 10-15% to the overall dataset.”

Reviewer #3 (Remarks to the Author):

This study provides a comprehensive anatomical description of two-dimensional orbit shape in Mesozoic archosaurs through geometric morphometrics. From my perspective, the study is largely confirmatory in that it corroborates patterns that have emerged through detailed anatomical descriptions. I offer a few major comments:

1) The study is almost entirely descriptive and is not driven by a specific hypothesis or question. It strikes me as somewhat of a “fishing expedition” even though there are existing hypotheses in the literature that could be tested. For example, the hypothesis that elliptical orbit shapes increase skull strength --- that hypothesis represents an opportunity for a detailed analysis. The study offers some data in support of this hypothesis, but the sample size is small. I would also strongly suggest adding more details on muscle anatomy, insertion points to make the models even better; there is much potential for improvement.

Lautenschlager: With all due respect, but did the reviewer read the manuscript? Firstly, could you please explain how the study is descriptive (beyond the necessity of describing/explaining the results)? The study is centred around a range of quantitative analyses to characterise the differences in orbit shape and their distribution across phylogeny and time and subsequently tests the biomechanical behaviour of these different orbit shapes in a variety of scenarios. This investigation into the form-function relationships has been made clear in the original text: “Here, I used geometric morphometric analysis (GMM) to characterise orbit shape across Archosauromorpha and throughout the Mesozoic to quantify the morphological diversity and changes thereof through time...Results from the shape analysis were used for the generation of different theoretical models subsequently subjected to biomechanical analysis to test the functional significance of specific orbit shapes.” Furthermore, the hypotheses referred to by the reviewer have been discussed and tested (even if this has not been bluntly phrased in the introduction) in the discussion: “Previous studies have hypothesised that orbit shape was related to feeding biomechanics rather than visual aspects and specifically that elliptical orbits correlated with increased skull strength^{22,23}. This hypothesis has been confirmed here in that elliptical, wedge- and keyhole-shaped orbits recorded lower stresses (Fig. 5) and deformation (Fig. 6) in the tested hypothetical models.”

Secondly, I cannot see how the sample size is small. The GMM data set consists of over 400 specimens (the maximum number of samples considering the preservation of fossil material). The subsequent biomechanical analyses consist of 64 + 5 + 2 different models/scenarios – substantially larger than previous analyses on the subject. Both Chure (1998) and Henderson (2003), the most comprehensive works on these groups, included 17 specimens. This is by no means meant to demean the work of the authors on two great studies but I am a bit baffled by the reviewer’s comment on small sample sizes.

With regards to the *Tyrannosaurus* model, I have now done a full muscle reconstruction for the species/model based on previously published methods and re-run the analyses with the more realistic muscle values. Supplementary figure 5 has been added showing the relevant details for the muscle reconstruction.

2) An important point to consider is that the “extravagant orbit anatomy” may not have a direct, or immediate function at all, like Gould’s spandrels. It is well known from mammals and birds that the eye gets proportionately smaller with eye socket size. In large skulls never fills the orbit completely. This known correlation renders the model in Fig 7b, with a huge eye filling all of the orbit as non-sensible. I would emphasize here again that it would be necessary to carry out more detailed

reconstructions of jaw musculature, eyes and eye muscles to better figure out the “soft” anatomy in this region.

Lautenschlager: It is correct that not every feature observed in modern or fossil organisms must have an immediate function. That is exactly why this study tests if the extravagant orbits convey any advantage. Based on the presented results the answer is a confirmation that they do. Biomechanical techniques are well-suited to address such questions and hypotheses as they provide quantifiable results.

It is further correct that orbit shape does not necessarily predict eye size. However, this is not inferred or suggested here. Neither is it the focus of this study to do so with the exception of the model shown in figure 8b. As stated, this is a hypothetical model intended to show the extreme end of a spectrum showing that the large eyes are physiologically and functionally disadvantageous. That said, I have now done a full muscle reconstruction for the species/model based on previously published methods and re-run the analyses with the more realistic muscle values. Supplementary figure 5 has been added showing the relevant details for the muscle reconstruction.

3) The 2-dimensional nature of the data is also problematic as the orientation of eye sockets varies substantially across archosaurs. Flattening all orbits into 2 dimensions will distort shapes and can bias the pattern. In this context it should also be noted that the orientation of the eye socket in itself is perhaps more relevant than details of the anatomy of the socket margin itself, as the in-depth studies of mammalian orbits have demonstrated convincingly.

Lautenschlager: Of course having data for 3D-orbit shape and orientation would likely improve the resolution but collecting this data for the same amount of specimens that are distributed in collections across the globe is not easily achieved. However, in contrast to mammals, the orientation of the orbit is very much lateral in archosauromorphs so that the 2D approach taken here is a good proxy. I agree that this could not necessarily be done for most mammals (modern or fossil). Furthermore, this problem is addressed by using different sets of functional analysis which test the problem in different 2D AND 3D settings.

4) The lack of a phylogenetic framework of the analysis is also problematic. It is very widely known that phylogeny is one of the major factors influencing anatomy, and there are many different techniques that can account for phylogenetic covariance, including for geometric morphometrics and PCA. Phylogenies are available and can be time calibrated. Adding phylogenetic context also facilitates much more advanced possibilities to illustrate character evolution.

Lautenschlager: I agree. A comprehensive phylogeny has been created for the sampled taxa and orbit shape has been mapped onto this phylogenetic tree to outline the occurrence of non-circular orbit shapes across the studied archosauromorphs. Details are provided in the methods (lines 263-273) and the results (lines 107-111), as well as illustrated in the new figure 3.

Reviewers' comments:

Reviewer #1 (Remarks to the Author):

I want to thank the author to address all the comments I made in the previous round. I am pleased with the answers and clarifications. However, there is only one point that I think that has not been addressed properly: the answer of my question number two about the influence of the morphologies in the stress results. I wrote an additional report explaining my concerns.

Review – second round

I want to thank the author to address all the comments I made in the previous round. I am pleased with the answers and clarifications. However, there is only one point that I think that has not been addressed properly: the answer of my question number two about the influence of the morphologies in the stress results. I am open to discuss this with the author directly, so he can contact me at jordi.marce@urv.cat if he has doubts, questions or does not agree with my approach. I will be happy to discuss it with him.

I am not sure that the results obtained in the first mechanical case are reliable and reflect the problem the author is facing. Due to this I run four different tests with a squared geometry of 1 mm of thickness in four different side sizes:

- 1) 9mmx9mm square with a circle of 8 mm diameter
- 2) 10mmx10mm square with a circle of 8 mm diameter
- 3) 20mmx20mm square with a circle of 8 mm diameter
- 4) 10mmx10mm square but with a circle of 7 mm diameter

I applied a distributed force in both upper and lower side to simulate a pure traction. The value of the force is according to what the author is doing now with the scaling method.

Then, I created a hole inside the square in the four cases. The hole has the same size in three cases: circle of 8 mm diameter and different size in one case: 7 mm. Applying the scaled force using what the author is proposing: the quasi-homothetic transformation for plane stress published in Marcé-Nogué et al 2013. For each case, the value of the force to apply is:

CASE	Square surface	Circle surface	Total Surface	Thickness	Scaled Force
1	81 mm ²	16π mm ²	30.73 mm ²	1 mm	8.68 N
2 (reference)	100 mm ²	16π mm ²	40.73 mm ²	1 mm	10 N
3	400 mm ²	16π mm ²	349.73 mm ²	1 mm	29.30 N
4	100 mm ²	14π mm ²	56.018 mm ²	1 mm	11.73 N

And running a FEA analysis with a convergence test of the value under study for having an adequate mesh:

I have three different stress values with the same circular morphology and the same size because the results of stress depend on the space we have between the hole and the wall of the square and not because the morphology of the hole. Therefore, the effects of orbit shape cannot be studied if this effect is not considered. The last case is the same morphology but different orbital size and with the same plate. The effect of the size is not removed due to the scaling applied in the force because the results are different.

Moreover, it is obvious that the average values of stress will change depending on the size of the plate irrespective of the orbital morphology. I wonder how this situation can affect the results of the mechanistic models of the theoretical shapes. Is the author choosing arbitrarily a size of the plate? Can the author reflect reality selecting different plate sizes?

My suggestion is not using average stress, rethink the size of the plates and include another parameter that can be of great interest in these known situations, the **stress concentration factor**: https://en.wikipedia.org/wiki/Stress_concentration

Reviewer #1

The presented manuscript tested some functional hypothesis of the orbits in archosaurs that can be I want to thank the author to address all the comments I made in the previous round. I am pleased with the answers and clarifications. However, there is only one point that I think that has not been addressed properly: the answer of my question number two about the influence of the morphologies in the stress results. I am open to discuss this with the author directly, so he can contact me at jordi.marce@urv.cat if he has doubts, questions or does not agree with my approach. I will be happy to discuss it with him.

I am not sure that the results obtained in the first mechanical case are reliable and reflect the problem the author is facing. Due to this I run four different tests with a squared geometry of 1 mm of thickness in four different side sizes:

- 1) 9mmx9mm square with a circle of 8 mm diameter
- 2) 10mmx10mm square with a circle of 8 mm diameter
- 3) 20mmx20mm square with a circle of 8 mm diameter
- 4) 10mmx10mm square but with a circle of 7 mm diameter

I applied a distributed force in both upper and lower side to simulate a pure traction. The value of the force is according to what the author is doing now with the scaling method.

Then, I created a hole inside the square in the four cases. The hole has the same size in three cases: circle of 8 mm diameter and different size in one case: 7 mm. Applying the scaled force using what the author is proposing: the quasi-homothetic transformation for plane stress published in Marcé-Nogué et al 2013. For each case, the value of the force to apply is:

And running a FEA analysis with a convergence test of the value under study for having an adequate mesh:

I have three different stress values with the same circular morphology and the same size because the results of stress depend on the space we have between the hole and the wall of the square and not because the morphology of the hole. Therefore, the effects of orbit shape cannot be studied if this effect is not considered. The last case is the same morphology but different orbital size and with the same plate. The effect of the size is not removed due to the scaling applied in the force because the results are different.

Moreover, it is obvious that the average values of stress will change depending on the size of the plate irrespective of the orbital morphology. I wonder how this situation can affect the results of the mechanical models of the theoretical shapes. Is the author choosing arbitrarily a size of the plate? Can the author reflect reality selecting different plate sizes?

My suggestion is not using average stress, rethink the size of the plates and include another parameter that can be of great interest in these known situations, the stress concentration factor: https://en.wikipedia.org/wiki/Stress_concentration

Lautenschlager: I would like to thank Dr Marcé-Nogué for taking the time to test this and for being available for discussion to find a suitable metric for the FEA models. Following his suggestions and discussion the stress concentration factor is now used to describe the performance of the different theoretical models. Updated results are presented in figures 6a-c. The new results do not change the overall interpretation. In fact, they do show an even stronger indication for a stress reduction function of the non-circular orbit shapes.

Additional text has been added to the manuscript to update the results and to provide an explanation of the stress concentration factor in the methods (lines 120-125; 346-355):

“A series of different biomechanical analyses were performed to assess the functional impact of different orbit shapes. Subjecting simplified planar models to different compressive and shear scenarios demonstrates that a deviation from the circular orbit morphology can be beneficial in reducing stress concentration under these conditions (Fig. 6). In all tested scenarios simulating dorsoventral and anteroposterior compression and dorsal and anterior shear, keyhole-shaped orbit models experience reduced stress compared to more circular models (Fig. 6a-d). ”

“All models were imported into Abaqus (Simulia, v. 6.141) for analysis and post-processing. Biomechanical performance for the flat planar models was assessed by calculating a stress concentration factor. Stress concentrations are formed by the presence of discontinuities (e.g. orbit shapes) in a structure which would otherwise show a uniform stress distribution⁶². A stress concentration factor is a dimensionless metric describing the ratio between peak stress to a reference stress and therefore describes how stress magnitudes are increased due to the introduction to a discontinuity. The reference stress was calculated from the normal stress for the dorsal and anterior compression models and as maximum normal stress for the dorsal and anterior shear models. Peak stresses were obtained from the FE models along the margin or the orbit shape (thereby avoiding artificially high stresses at the constrained nodes).”